

# Higher-Order Calibration on WindRAD scatterometer winds

Zhen Li[1], Ad Stoffelen[1], and Anton Verhoef[1]

[1]Utrechtseweg 297, 3731 GA De Bilt, the Netherlands

**Correspondence:** Z. Li (zhen.li@knmi.nl)

**Abstract.** WindRAD (Wind Radar) is a dual-frequency rotating fan-beam scatterometer instrument on the FY-3E (Fengyun-3E) satellite. Scatterometers are generally calibrated using the linear NOC (NWP Ocean Calibration) method, to control the main gain factor of the radar. While WindRad is stable, the complex geometry, the design of the instrument, and the rotating antenna make the backscatter ($\sigma°$) distributions persistently non-linear, hence NOC is insufficient. Therefore, a higher-order

calibration method is proposed, called HOC. The CDF (Cumulative Distribution Function) matching technique is employed to match the CDF of measured $\sigma°$s to simulated $\sigma°$s. HOC removes the non-linearities for each incidence angle. However, it is not constructed to remove the anomalous harmonic azimuth dependencies caused by the antenna rotation. These azimuth dependencies are reduced by NOCant (NOC as a function of incidence angle and relative antenna azimuth angle). Therefore, the combination of HOC&NOCant is implemented to correct both anomalous $\sigma°$ amplitude and azimuth variations. The wind

retrieval performance is evaluated with NOCant, HOC, and HOC&NOCant combined. The wind statistics and the cone distance metric both show that HOC&NOCant achieves the optimal winds for C-band and Ku-band. The calibrations have been tested on two operational input data versions; HOC works well on both data versions and HOC&NOCant can achieve the optimal wind performance for both data versions. This confirms the usefulness of HOC calibration in the case of non-linear instrument gain anomalies.

## 15  1  Introduction

FY-3E (Fengyun-3E) is part of the Chinese FY-3 meteorological satellite series. It was launched by the China Meteorological Administration (CMA) on 5th July 2021 and carries the WindRAD (Wind Radar) scatterometer. It is in an early morning, near-polar, sun-synchronous orbit; the FY-3 satellite series including FY-3C, FY-3D, FY-3E, and other international meteorological satellites provide good coverage for the daily cycle, which is beneficial for numerical weather prediction (NWP), climate stud-

ies, and environmental science (Zhang et al. (2022)).

WindRAD is the first dual-frequency (C-band at 5.40 GHz and Ku-band at 13.256 GHz), four antennas (each frequency in both HH and VV polarization) rotating fan-beam scatterometer. Another rotating fan-beam scatterometer in orbit is CSCAT (CFOSAT Scatterometer) onboard CFOSAT (Chinese-French Oceanography Satellite) (Liu et al. (2020); Li et al. (2021)). The

main difference is that WindRAD is dual-frequency, whereas CSCAT is Ku-band only. The WindRAD data characteristics and wind retrieval performance are investigated in Li et al. (2023). The level-1 data are organized in WVCs (Wind Vector



Cells) with 70 WVCs of size 20 km × 20 km across the swath, which is a sub-track coordinate system with the axes oriented along and across the swath (Figure 1). The C-band and Ku-band antennas are installed at the two sides of the instrument as shown in Figure 2. The pulses emitted from the antenna beams sweep conically over the swath while the satellite flies forward, which leads to overlapping views in each WVC with a variety of geometries (incidence angles, azimuth angles, polarization, and frequencies). The distributions of the geometries are the most diverse in the sweet swath (the WVCs between the nadir and outer swath), the least diverse in the outer swath with only high incidence angles and azimuth angles around 90° and 270°, and with limited diversity in the nadir swath with only forward and backward facing azimuth angles. The more diverse, the better geometrically sampled for wind retrieval, whereas a limited diversity makes accurate wind retrieval more challenging.

In order to derive accurate winds from a scatterometer, empirical backscatter calibration is an essential step. A well-elaborated and widely used calibration method is NOC (NWP Ocean Calibration), which assesses the difference between the measured $\sigma°$s and the simulated $\sigma°$s from collocated model winds and corresponding GMF (Geophysical Model Function) (Stoffelen and Anderson (1997)). This method makes the measured data align with the GMF by using a corrected instrument gain value per incidence angle and polarization. It assists to derive accurate winds even if the original scatterometer gain value is inaccurate (Stofflen (1999)). The NOC method has been successfully implemented in various scatterometers, such as fixed fan-beam (Verspeek et al. (2012)), rotating pencil-beam (Yun et al. (2012)), and rotating fan-beam (Li et al. (2021)) instruments. Li et al. (2023) investigated different NOC methods for WindRAD: NOCinc (NOC as a function of incidence angle) and NOCant (NOC as a function of incidence angle and relative antenna azimuth angle). NOCant takes the rotating antenna into account by including the relative antenna azimuth angle in the calculation. The relative antenna azimuth angle is defined with respect to the satellite flight direction. It appears that the NOCinc correction is insufficient to correct the $\sigma°$ distributions because it sums up the corrections of all the relative antenna azimuth angles at a specific incidence angle, which averages the effects due to the rotating antenna. On top of that, the $\sigma°$ distribution of the rotating fan-beam system contains persistent non-linear effects which cannot be calibrated by the NOC method. In this paper, we propose a higher-order calibration for $\sigma°$s, called HOC, to calibrate the non-linear effect. HOC calibration can also be combined with NOC calibration, their results are discussed in this paper. Section 2 introduces the datasets used in this study and describes HOC calibration. The results of HOC calibration on the $\sigma°$s are shown in section 3. Wind retrieval assessment with HOC and NOC and discussion of the results are presented in section 4. Finally, the conclusions are presented in section 5.

## 2 Datasets and HOC calibration

### 2.1 Datasets

The WindRAD operational level-1 data contain C-band and Ku-band $\sigma°$s with both horizontal (HH) and vertical (VV) polarization (Li et al. (2023)). The incidence angle range for C-band is between 33.0° and 47.0°, and for Ku-band it is between 36.5° and 44.0°. The WVC size of 20 km × 20 km is selected (level-1 data also contain data on a 10 km grid but these are not used). There are two data versions: operational data 'v1oper' in the period from 15 March 2022 till 27 April 2023, and v2oper



data starting from 4 May 2023. The v1oper data from 1 Dec to 31 Dec 2022 are used to derive HOC calibration. In the v2oper data, some bugs in v1oper have been corrected. We have two test data sets of v2oper, one for C-band (15 Sept to 17 Sept 2022, 1 Dec to 7 Dec 2022) and one for Ku-band (1 Nov to 3 Nov 2022). HOC calibration is tested both on the v1oper and on the v2oper versions to confirm the consistency of the calibration method for different data versions.

The NWP (Numerical Wind Prediction) winds are derived from the ECMWF (European Centre for Medium-Range Weather Forecasts) operational forecast model. The hourly forecast steps of +3 h, +4 h, ..., and +21 h are used and the model winds have been interpolated to the WVC in time and location and neutral 10-m winds have been converted to stress-equivalent 10-m winds, correcting for air mass density. Model winds are appropriate for HOC calibration because of their availability at every scatterometer WVC. Their spatial representativeness and high quality have been investigated in detail in Vogelzang and
Stoffelen (2022) with a careful error assessment of scatterometer winds, in-situ winds, and ECMWF model stress-equivalent winds.

## 2.2  HOC Calibration

HOC uses the CDF (Cumulative Distribution Function) matching technique to calculate $\sigma^\circ$ dependent calibrations. This method has been applied to ERS for wind speed bias correction (Stoffelen (1998)), the GMF NSCAT for Ku-band (Wentz and Smith
(1999)), and the C-band GMF CMOD7 for eliminating the WVC dependence of the wind speed PDF (Probability Distribution Function) (Stoffelen et al. (2017)). HOC was also implemented for the intercalibration of backscatter measurements among the Chinese HY-2 (HaiYang-2) satellites, to achieve consistent backscatter measurements (Wang et al. (2021)).

In this study, we apply HOC calibration between the measured $\sigma^\circ$s and the simulated $\sigma^\circ$s, computed from the collocated
NWP winds through the corresponding GMF. Figure 3 illustrates the principle of CDF matching. The black curve is the reference CDF, whereas the grey curve is the CDF of the data to be calibrated with respect to the reference. To each uncalibrated data point $x$ a calibrated $\tilde{x}$ can be found, and the original CDF value at $x$ equals to the reference CDF value at $\tilde{x}$. $x$ here represents the measured $\sigma^\circ$, whereas $\tilde{x}$ represents the calibrated $\sigma^\circ$. We take the CDF of the simulated $\sigma^\circ$ data as a reference and the CDF of the measured $\sigma^\circ$ data is calibrated in dB unit space with respect to the reference (corresponding to a non-linear cali-
bration for the $\sigma^\circ$ in linear unit space). The measured $\sigma^\circ$ data are then calibrated (matched) statistically to the simulated $\sigma^\circ$ data.

To be clear, HOC is without doubt a physically useful method in the absence of uncertainty in all sources. However, uncertainty in the data and models may potentially deviate the expected physical measured and simulated distributions. As uncertainty is convolved with the underlying physical distribution, CDF mapping still makes physical sense when the uncertainty
in both measured and simulated $\sigma^\circ$ is similar. The main uncertainty in the simulated $\sigma^\circ$ is the uncertainty in the wind. It is well described by normal errors in both wind components and about 1 m/s (Vogelzang and Stoffelen (2022)). The GMF and other parameters contribute much less to the simulated $\sigma^\circ$ uncertainty. At 10 m/s this implies a 10 percent error, or 0.4 dB, and similarly ∼2 dB at 2 m/s. It is clear that HOC should work well for high winds in particular, where the uncertainties in the



measured and simulated $\sigma°$ match best, although the measurement uncertainty strongly depends on the incidence angle. At very

low $\sigma°$ and winds, both the simulated and measured $\sigma°$ values are mainly determined by uncertainty and the calibration will

be artificial. Note that the $\sigma°$ dependence on wind speed is quasi-linear, such that any HOC calibration error would manifest

itself in wind-speed dependent verification.

## 3   HOC results

HOC calculates $\sigma°$ dependent calibration in intervals of 0.1 dB or about 2 percent. The $\sigma°$ distribution has a dependency on

incidence angles (Li et al. (2023)), thus HOC is derived and implemented on $\sigma°$s as a function of incidence angle for C-band

and Ku-band (HH and VV polarization), respectively.

### 3.1   C-band

Fig. 4 shows the C-band HH contoured histogram of measured $\sigma°$ versus simulated $\sigma°$ for three different incidence angles.

C-band VV has a very similar pattern as C-band HH, hence it is not shown. The simulated $\sigma°$s are calculated with collocated

ECMWF winds through CMOD_HH for HH polarization (newly developed by Wang, private communication, 2023) and

CMOD7 for VV polarization (Stoffelen et al. (2017)). Fig. 4 (a) to (c) show the selected incidence angles 34°, 38°, 44°,

representing low, medium, and high incidence angles, respectively. Note that there is a double mode split in the high $\sigma°$ value

area for the low incidence angle (Fig. 4(a)), which is caused by the level-0 data processing, this issue has been corrected in

the v2oper data. All these $\sigma°$ distributions show non-linearity, especially at the low $\sigma°$ values, where asymmetries from the

diagonals occur. NOC calibration has been implemented (Li et al. (2023)) and shows that NOCant takes the azimuth variations

into account, yielding a better calibration result as compared to NOCinc. However, the non-linearities in the $\sigma°$ distributions

persist with NOC calibration. As described in section 2, HOC employs the CDF matching technique to match the CDF of the

measured $\sigma°$ to the simulated one. Fig. 5 shows the CDF of measured and simulated $\sigma°$ for a number of incidence angles. Due

to the larger deviation from the diagonal of the $\sigma°$ distribution at the low and high incidence angles, the low and high incidence

angles show larger differences between the CDFs. Their corresponding PDFs are shown in Fig. 6. After applying HOC, the

CDFs of measured $\sigma°$ are well matched with the simulated $\sigma°$ (Fig. 7 (a)), as well as the PDFs (Fig. 7 (b)). The asymmetry

of the $\sigma°$ distribution is corrected after HOC calibration (Fig. 4 (d) to (f)), which indicates the non-linear effect is successfully

removed empirically.

### 3.2   Ku-band

Fig. 8 (a) to (c) show the Ku-band VV measured $\sigma°$ versus simulated $\sigma°$ distributions at incidence angles of 37°, 39°, and 40°,

which correspond to low, medium, and high incidence angles, respectively. The simulated $\sigma°$s are calculated from collocated

ECMWF winds, using the NSCAT4-DS GMF (Wang et al. (2017)). The distribution of Ku-band HH polarization $\sigma°$s has a

similar pattern as for VV, hence it is not shown here. Note that, similar to C-band, there is a double mode split in the high

$\sigma°$ value area for the low and high incidence angles (Fig. 8(a), (c)), which is also caused by the level-0 data processing. This





has been corrected in the v2oper data. Contrary to the C-band results, the non-linear distribution at the medium incidence angle is not so obvious, whereas it is pronounced at the low and high incidence angles. This asymmetry cannot be corrected by NOCinc or NOCant (Li et al. (2023)), therefore, HOC calibration is implemented. The CDF and PDF distributions of the selected incidence angles are shown in Fig. 9 and 10, the largest mismatch occurs at the low and high incidence angles. After HOC calibration, the CDFs and PDFs match well with each other (Fig. 11) and the $\sigma^\circ$ distributions are much more symmetric

(Fig. 8 (d) to (e)).

## 4   Wind retrieval assessment and discussion

The most widely used wind inversion algorithm is the so-called MLE (Maximum Likelihood Estimation) method, which is based on a Bayesian approach (Chi and Li (1988); Pierson (1989); Portabella and Stoffelen (2002); Cornford et al. (2004); Stoffelen and Portabella (2006)). Here is a brief summary of this method, the details of the inversion method can be found in

Stoffelen and Anderson (1997); Vogelzang and Stoffelen (2018).The MLE function is defined for each WVC as:

$$MLE = \frac{1}{N}\sum_{i=1}^{N}(\frac{\sigma^\circ_{mi} - \sigma^\circ_{si}}{K_p(\sigma^\circ_{xi})})^2, \tag{1}$$

$N$ is the number of views in the WVC, $\sigma^\circ_{mi}$ is the measured $\sigma^\circ$, $\sigma^\circ_{si}$ is the simulated $\sigma^\circ$, $K_p(\sigma^\circ_{xi})$ is the expected Gaussian observation noise with the form of $K_p \times \sigma^\circ_{xi}$, and $\sigma^\circ_{xi}$ is usually taken to be either $\sigma^\circ_{mi}$ or $\sigma^\circ_{si}$. $K_p$ is the measurement error variance determined by instrument noise. With known incidence and azimuth angle, $\sigma^\circ_{si}$ is related to wind speed and wind

direction and derived through a GMF. The goal is to minimize the cost function (1) using different wind speed and direction trial values. The trial value that yields the lowest MLE is the retrieved wind vector.

As discussed in Li et al. (2023), the $\sigma^\circ$s at high and low incidence angles are more severely influenced by noise and other factors, hence only the incidence angles between 36° and 43° are used for C-band wind retrievals, whereas only the incidence

angles between 38° and 41° are used for Ku-band.

### 4.1   C-band wind statistics and discussion

The C-band wind retrievals have been performed using MSS (Multiple Solution Scheme, Portabella (2002)) and 2DVAR (Two-Dimensional Variational Removal, Vogelzang et al. (2009)). C-band is hardly influenced by rain, hence rain contamination QC (Quality Control) is not used in the test. HOC, HOC&NOCinc, and HOC&NOCant calibrations are assessed and discussed in

this section.

The radar antenna gain is strongly incidence angle dependent, and the linear gain value is the prime uncertainty parameter in many scatterometers (Verspeek et al. (2012), Yun et al. (2012), Li et al. (2021)). For WindRAD we also observe a non-linear gain as described in section 3.1. In this section, we test if HOC can also correct the incidence angle dependence (NOCinc) and



the rotation azimuth-angle dependence (NOCant).

Fig. 12a shows the NOCinc calculated with the original $\sigma°$s, whereas Fig. 12b shows the NOCinc calculated with the HOC-calibrated $\sigma°$s. The NOCinc after HOC calibration is almost flat except at the highest incidence angle, whereas the original NOCinc contains high calibration values at both low and high incidence angles. This indicates that HOC actually also cor-

rects the incidence angle dependencies. Note that the vertical scale of Fig. 12b is much smaller than in Fig. 12a. The wind retrieval using HOC&NOCinc has been tested, and the retrieved wind statistics give a very similar result for HOC-only and HOC&NOCinc (not shown). This implies that HOC is able to correct the incidence angle dependencies and the non-linear gain, hence the combination of HOC&NOCinc has a similar effect as HOC-only, as expected.

NOCant takes the relative antenna azimuth angle into account, aiming to reduce the azimuth-dependent backscatter variations. In Fig. 13 (a) and (b), NOCant corrections derived from the original $\sigma°$s are shown, whereas in (c) and (d) NOCant corrections derived from the HOC-calibrated $\sigma°$s are shown. Unlike the case where NOCinc is applied after HOC, the azimuth-dependent variations remain after HOC calibration, which is a consequence of that HOC is not set to correct the azimuth angle dependencies. However, the amplitude of the NOCant correction somewhat reduces after HOC, which is probably caused by

HOC calibration as a function of $\sigma°$ interfering with the errors in the simulated $\sigma°$s. Therefore, we expect the wind retrieval can be further improved with the combination of HOC&NOCant.

The wind retrieval statistics using NOCant, HOC, and HOC&NOCant are compared. Fig. 14 shows the wind speed/direction bias with respect to the ECMWF winds and their corresponding SDD (Standard Deviation Difference), as a function of

speed/direction. The wind speed bias using only NOCant increases from a negative to a positive bias with increasing wind speed between 0 m/s and 25 m/s. The wind speed biases are reduced when using either HOC or HOC&NOCant, with only a small positive bias at high wind speeds remaining. The wind direction biases are quite similar for all three calibration methods, whereas the wind direction SDD is reduced when using HOC&NOCant. Fig. 15 shows the wind statistics as a function of WVC across the swath. The lowest wind speed bias and SDD are obtained with HOC&NOCant, followed by HOC and

NOCant, respectively. For the wind direction bias and its SDD, HOC also shows larger values than NOCant. This is because HOC does effectively not remove the azimuth dependencies as we discussed in the last paragraph. Hence, HOC-only cannot achieve the best calibration.

The MLE (also called cone distance) from equation (1) is another metric to measure the quality of the retrieval. The MLE

reveals how well the measurements fit the GMF, hence the smaller the MLE, the better the measurements fit the GMF, and the better wind retrieval can be expected. The MLE is normalized by a WVC-dependent factor to obtain an expectation value of 1. This makes monitoring and quality control easier. Different MLE normalizations lead to different outcomes, therefore the same normalization and threshold are applied for all the calibration methods, to make the results directly comparable. Fig. 16 shows the normalized MLE per WVC number for NOCant, HOC, and HOC&NOCant. Obviously, HOC&NOCant achieves





the lowest MLE values across the swath, whereas HOC results in quite high values in the sweet swath. As we mentioned before, HOC does effectively not remove the antenna azimuth-dependencies, and the high MLE values result from too large deviations of $\sigma^\circ$ values from diverse azimuth views (Fig. 13), indicating anomalous azimuth dependencies. Overall, the best wind retrieval statistics and lowest MLE values for C-band are derived by the HOC&NOCant calibration.

**4.2   Ku-band wind statistics and discussion**

Ku-band wind retrievals have been performed using MSS (Portabella (2002)), SST corrections (Wang et al. (2017)), QC to remove rain contamination for Ku-band (Portabella and Stoffelen (2001)), and 2DVAR ambiguity removal (Vogelzang et al. (2009)).

Similar to C-band, HOC corrects not only the non-linearities in the $\sigma^\circ$ distributions but also the incidence angle dependencies, which makes the NOCinc after HOC-calibration flat for all incidence angles, except for the highest incidence angle (Fig. 17). This leads to similar wind retrieval performance for HOC-only and HOC&NOCinc. But HOC does also here not effectively correct the azimuth dependencies caused by the antenna rotation, hence the azimuth dependencies remain after HOC calibration, although with reduced amplitude (Fig. 18).


As discussed in section 4.1, we compare the wind retrieval statistics of NOCant, HOC, and HOC&NOCant for Ku-band as well. Fig. 19 shows the wind speed/direction bias and their corresponding SDD as a function of speed/direction. The wind speed bias with NOCant is quite high at wind speeds above 15 m/s. HOC calibration is able to reduce this wind speed bias above 15 m/s, and HOC&NOCant further reduces the wind speed bias and its SDD for all wind speeds. The average wind
speed bias as a function of WVC number (Fig. 20(a)) is smallest using HOC, but HOC shows the largest bias range among the three calibration methods, whereas HOC&NOCant shows the flattest bias. The use of HOC&NOCant also reduces the wind direction SDD (Fig. 20(d)). Similar to section 4.1, the MLE metric is tested for Ku band (Fig. 21). The MLE of HOC&NOCant is generally the lowest across the swath, which indicates the best fit to the GMF, but the difference between NOCant and HOC&NOCant is smaller than for C-band, probably because the non-linearity in the $\sigma^\circ$ distribution for medium incidence
angles (the used angles are 38° to 41°, Fig. 8) is not as large for Ku-band as for C-band (Fig. 4). Using HOC, the azimuth-angle dependencies cannot be removed, hence the highest MLE values are located at the sweet swath, corresponding to the highest deviations in the NOCant corrections (Fig. 18). In conclusion, we also consider the combination of HOC&NOCant as the optimal $\sigma^\circ$ calibration for Ku band, resulting in the best wind retrieval performance.

**4.3   HOC and HOC&NOCant consistency test with different data version**

There is a mirror effect between ascending and descending orbits for the v1oper data, which can be seen from the ascending and descending NOCant (Fig. 22). There is also a double mode split for high $\sigma^\circ$ values in the $\sigma^\circ$ distribution for low and high incidence angles (Fig. 4 and 8). These issues have been corrected in the v2oper data. NOCant, HOC, and HOC&NOCant have





been tested on the test data sets of v2oper. HOC&NOCant gives the best wind retrieval performance, which is consistent with the results from the v1oper data. The MLE metric is derived and the lowest MLE is obtained using HOC&NOCant for both

C-band and Ku-band. The sweet swath MLE peaks for HOC are not as big as for the v1oper data, but the MLEs show rather similar patterns to the corresponding NOCant and HOC&NOCant, which is mainly caused by the version change. It appears that after fixing the two input data issues mentioned at the beginning of this section, the azimuth dependencies, which influence the MLE derived from HOC strongest, are smaller than in the v1oper data. Overall, the wind retrievals from v2oper are better than those from v1oper, yielding better wind statistics. HOC and HOC&NOCant give consistent wind retrieval performance

for the different data versions.

## 5   Conclusions

WindRAD is the first dual-frequency rotating fan-beam scatterometer in orbit and its performance appears stable. A well-elaborated and widely used calibration method, NOC (including NOCinc and NOCant), has been investigated extensively for

WindRAD in Li et al. (2023). However, the complex geometry, the design of the instrument, and the rotating antenna induce persistent non-linearities in the $\sigma°$, even after NOC calibration. A higher-order calibration called HOC is proposed in this paper. It employs the CDF matching technique to calculate $\sigma°$ dependent calibrations per incidence angle, which matches the CDF of the measured $\sigma°$s to the simulated $\sigma°$s in dB space with an interval of 0.1 dB. It corresponds to a non-linear calibration for $\sigma°$ in linear space.


When NOCinc is computed after HOC is applied, the calibrations are close to zero (except for the highest incidence angle) for both C-band and Ku-band, which implies that HOC is able to calibrate at all incidence angles. Hence, it is not necessary to combine HOC&NOCinc. However, the combination of HOC&NOCant can further improve the wind retrieval performance. NOCant is able to further reduce the antenna azimuth dependencies. HOC in itself is not effectively set up to remove the

azimuth dependency. Thus, HOC corrects the non-linearities and the incidence angle dependencies, whereas NOCant corrects the antenna azimuth dependencies. It is indeed verified in this paper, that in this way the combination HOC&NOCant achieves optimal wind retrieval.

The wind retrieval results of NOCant, HOC, and HOC&NOCant are analyzed and compared. The MLE metric shows that

HOC&NOCant yields the best fit to the corresponding GMF for both C-band and Ku-band. For Ku-band, the MLE of NOCant and HOC&NOCant is much closer than for C-band because the used incidence angles are in the medium range, where the non-linearity at Ku band is less than at C band. Overall, HOC&NOCant gives the optimal wind speed/direction bias and lowest SDD values.



255 In the v1oper input data, a mirror effect of NOCant between ascending and descending orbits, and a double mode split for high values in the $\sigma^\circ$ distribution for low and high incidence angles are observed. These issues have been corrected in the v2oper data. HOC and HOC&NOCant have been tested on the test data sets of v2oper as well. The wind statistics for NOCant, HOC, and HOC&NOCant have been assessed. HOC&NOCant achieves the optimal wind retrieval result for the test data sets of v2oper as well, which is consistent with the result from the v1oper data.


 In summary, HOC can calibrate the non-linearity in the $\sigma^\circ$ distribution and the incidence angle dependency, but it cannot effectively remove the azimuth dependency. In combination with NOCant, which is designed to remove the azimuth dependency, HOC&NOCant achieves optimal wind retrieval performance. This method works not only on the v1oper data but also on the v2oper data, showing consistent results with different data versions. As the retrieval methodology and the GMFs used are

265 generic for all scatterometers in the virtual scatterometer constellation, this empirical intercalibration results can be employed to provide intercalibrated global user products for global user benefit.





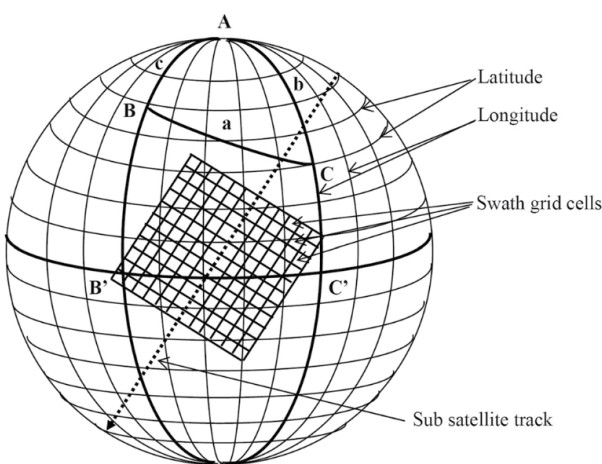

**Figure 1.** WVC coordinate system illustration, each grid cell represents one WVC (SCAT-DP team (2010)).

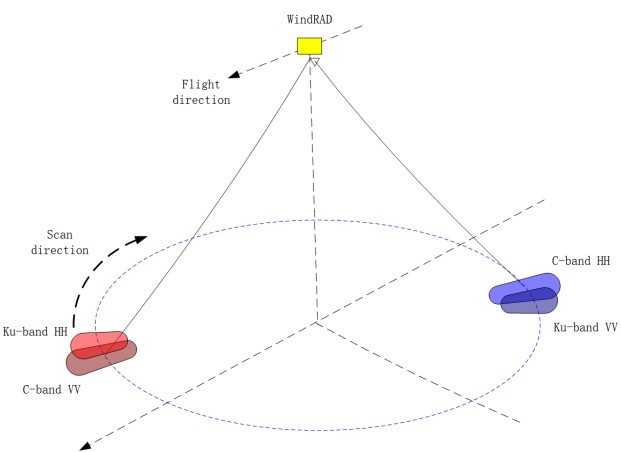

**Figure 2.** WindRAD scatteromter geometry (Li et al. (2023)).





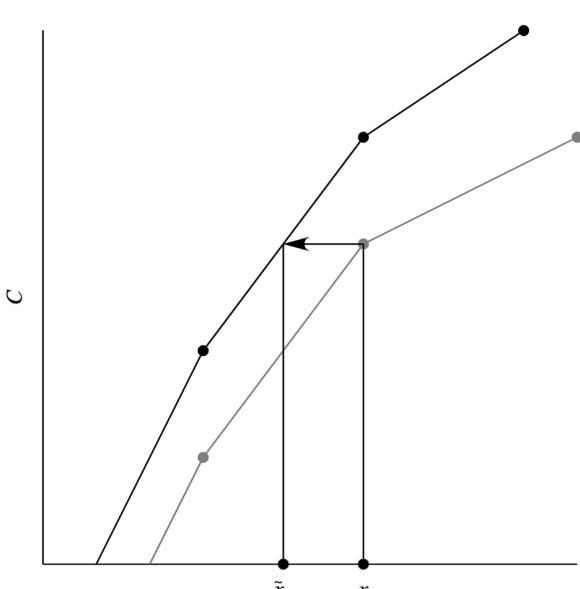

**Figure 3.** CDF matching technique illustration.





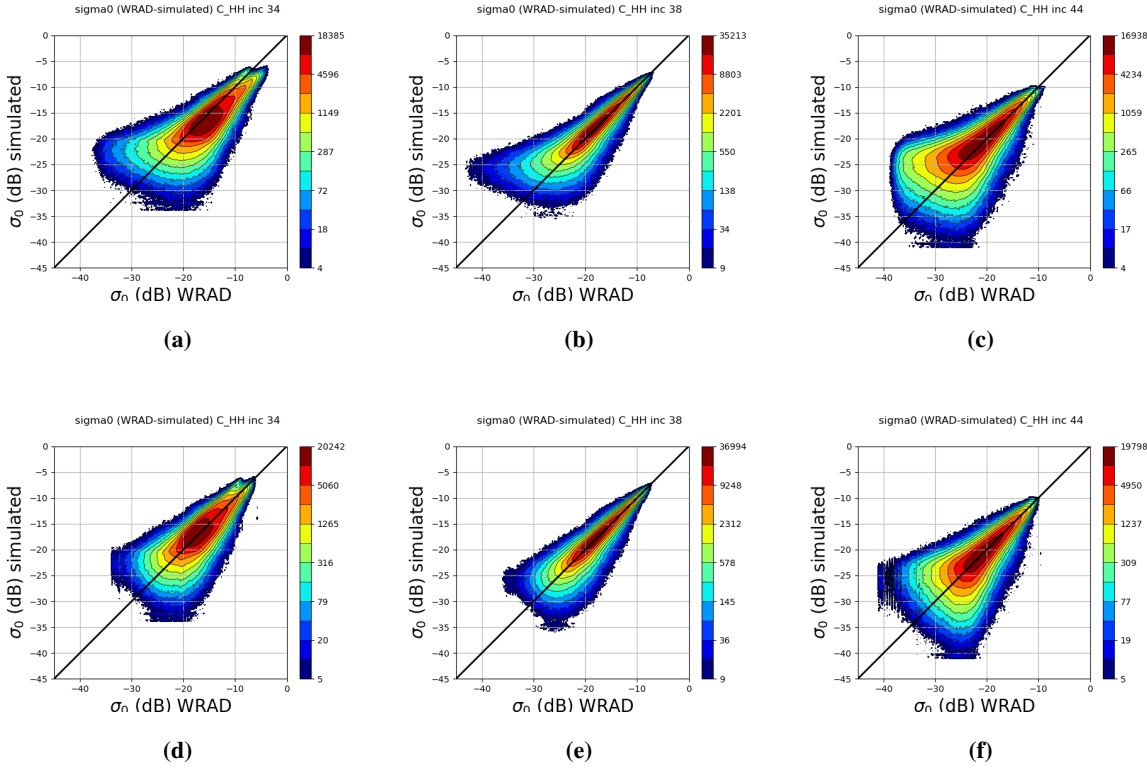

**Figure 4.** C-band HH polarization, measured $\sigma^\circ$ and simulated $\sigma^\circ$ joined distribution per incidence angle, data from 1 Dec - 31 Dec 2022 ascending orbits, upper panel is original distribution, the lower panel is HOC calibrated: (a) original incidence $34^\circ$, (b) original incidence $38^\circ$, (c) original incidence $44^\circ$, (d) HOC calibrated incidence $34^\circ$, (e) HOC calibrated incidence $38^\circ$, (f) HOC calibrated incidence $44^\circ$.

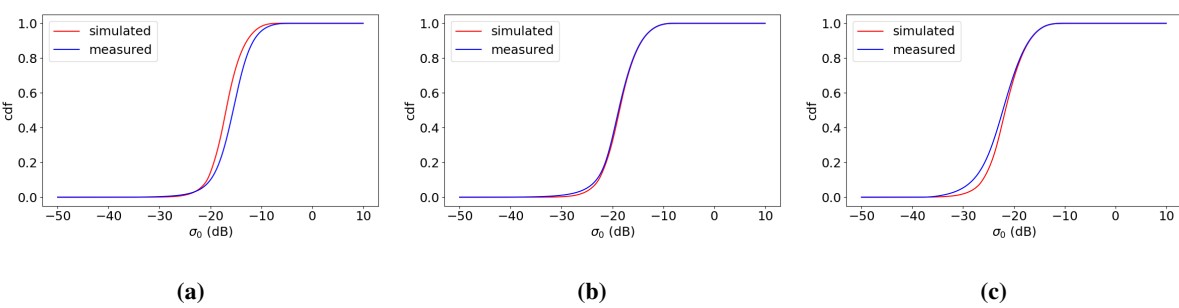

**Figure 5.** C-band measured $\sigma^\circ$ and simulated $\sigma^\circ$ CDF distribution per incidence angle, data from 1 Dec - 31 Dec 2022 ascending orbits: (a) HH incidence $34^\circ$, (b) HH incidence $38^\circ$, (c) HH incidence $44^\circ$.





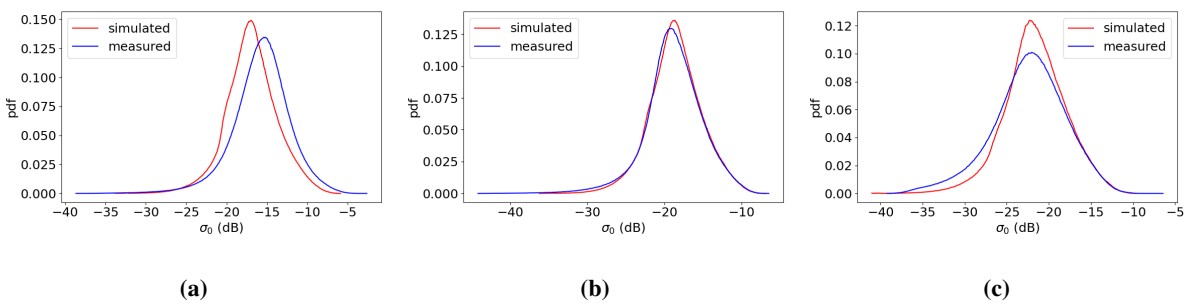

**Figure 6.** C-band measured $\sigma^\circ$ and simulated $\sigma^\circ$ PDF distribution per incidence angle, data from 1 Dec - 31 Dec 2022 ascending orbits: (a) HH incidence $34^\circ$, (b) HH incidence $38^\circ$, (c) HH incidence $44^\circ$.

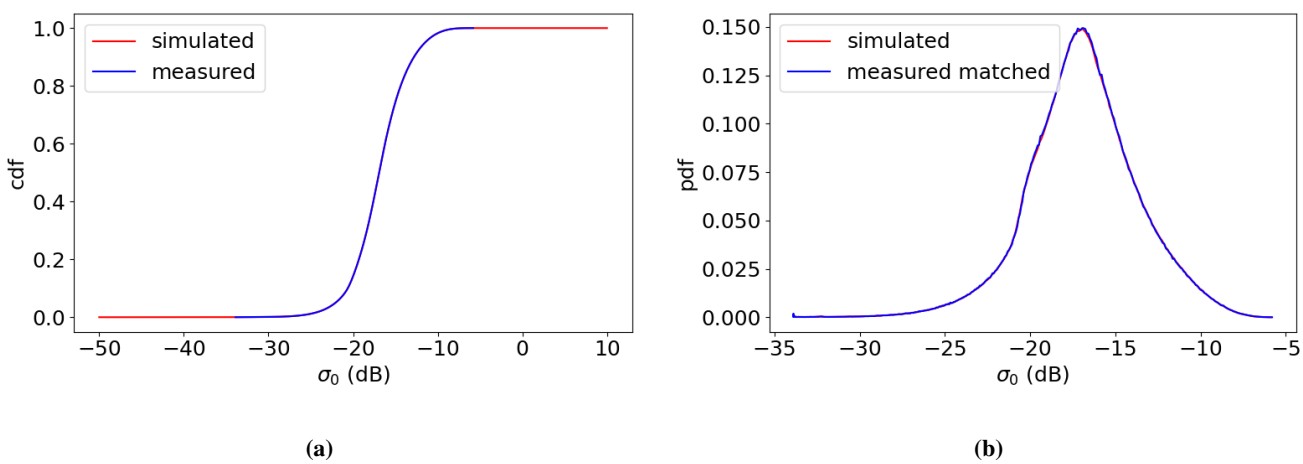

**Figure 7.** C-band HH measured $\sigma^\circ$ and simulated $\sigma^\circ$ matched CDF and PDF distribution at incidence $34^\circ$ (the other incidence angles are the same pattern, thus not shown here), data from 1 Dec - 31 Dec 2022 ascending orbits, (a) matched CDF; (b) matched PDF.





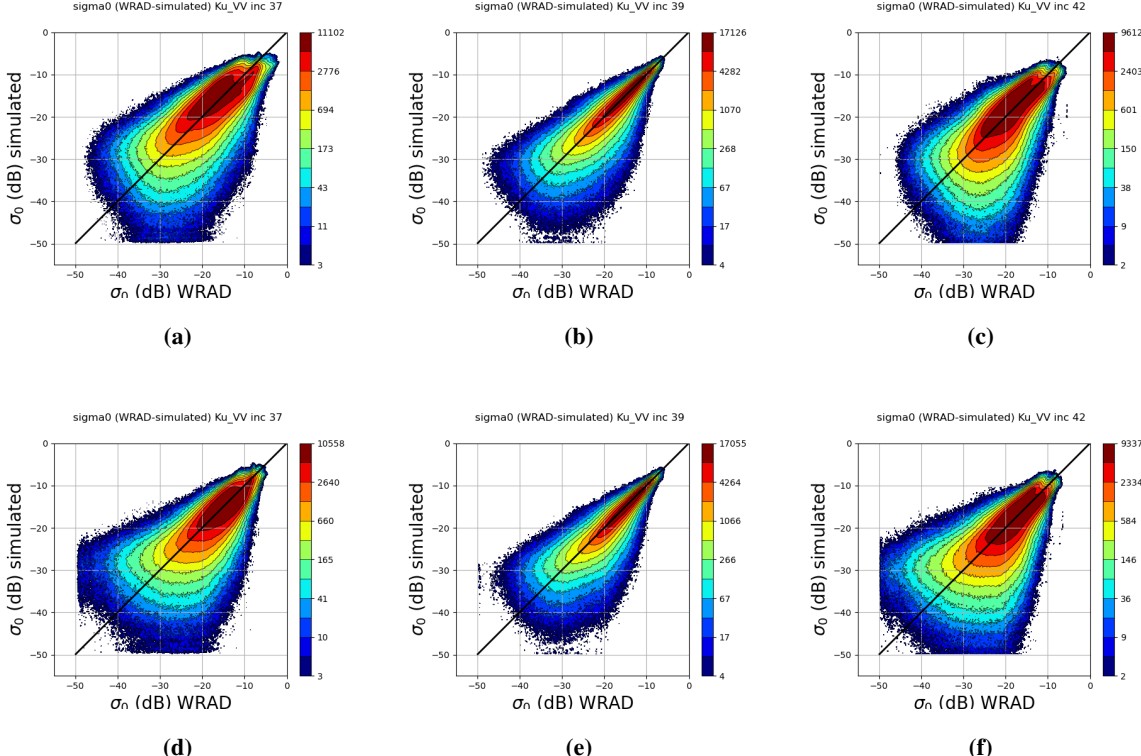

**Figure 8.** K-band VV polarization, measured $\sigma^\circ$ and simulated $\sigma^\circ$ joined distribution per incidence angle, data from 1 Dec - 31 Dec 2022 ascending orbits, upper panel is original distribution, the lower panel is HOC calibrated: (a) original at incidence $37^\circ$, (b) original at incidence $39^\circ$, (c) original at incidence $42^\circ$, (d) HOC calibrated at incidence $37^\circ$, (e) HOC calibrated at incidence $39^\circ$, (f) HOC calibrated at incidence $42^\circ$.

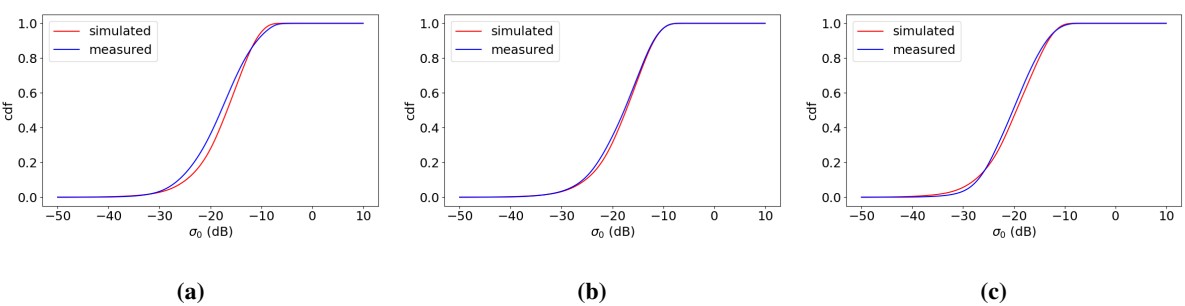

**Figure 9.** Ku-band measured $\sigma^\circ$ and simulated $\sigma^\circ$ CDF distribution per incidence angle, data from 1 Dec - 31 Dec 2022 ascending orbits: (a) VV at incidence $37^\circ$, (b) VV at incidence $39^\circ$, (c) VV at incidence $42^\circ$.





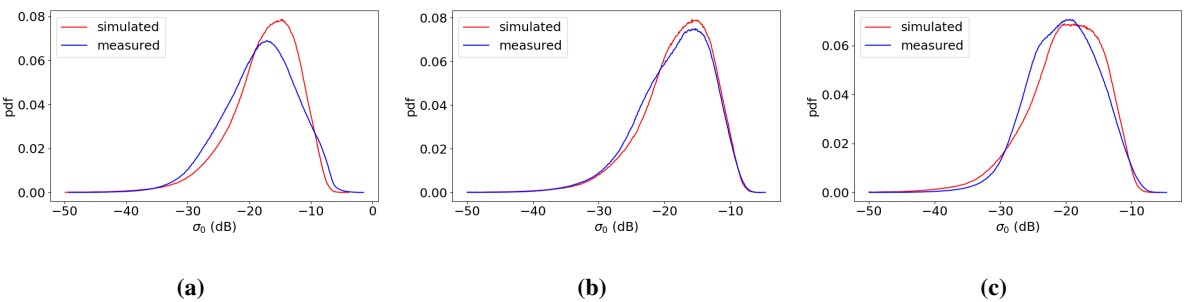

**Figure 10.** Ku-band measured $\sigma^\circ$ and simulated $\sigma^\circ$ PDF distribution per incidence angle, data from 1 Dec - 31 Dec 2022 ascending orbits: (a) VV at incidence $37^\circ$, (b) VV at incidence $39^\circ$, (c) VV at incidence $42^\circ$.

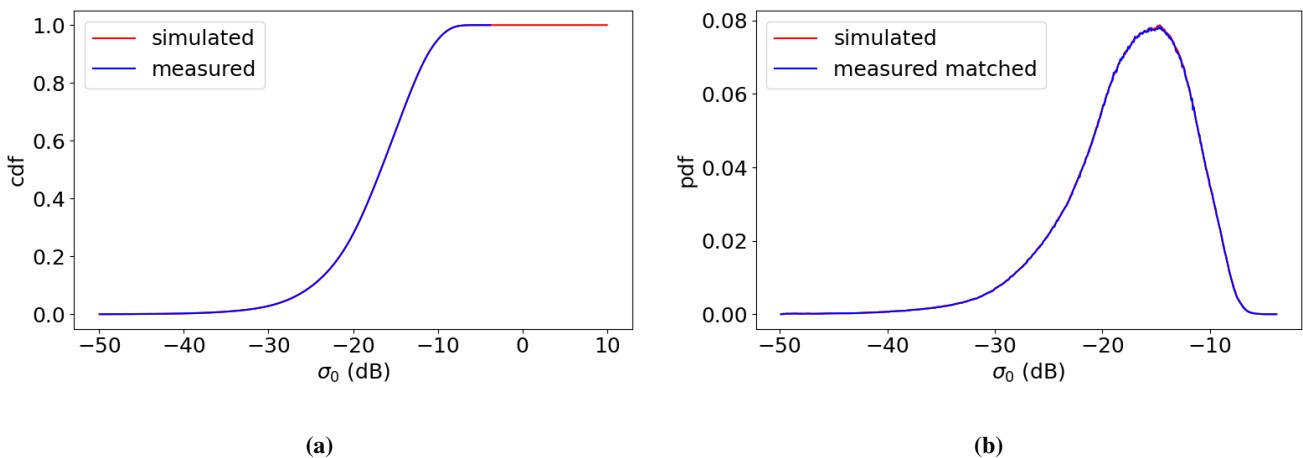

**Figure 11.** Ku-band VV measured $\sigma^\circ$ and simulated $\sigma^\circ$ matched CDF and PDF distribution at incidence $37^\circ$ (the other incidence angles are the same pattern, thus not shown here), data from 1 Dec - 31 Dec 2022 ascending orbits, (a) matched CDF; (b) matched PDF.

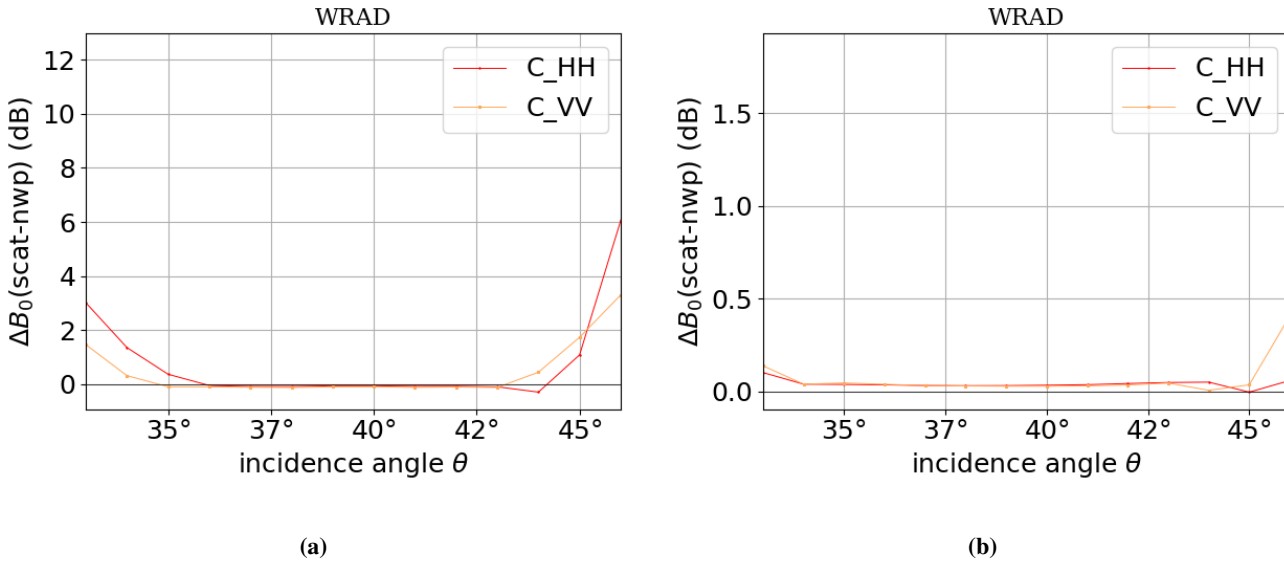

(a)                                                        (b)

**Figure 12.** C-band NOCinc (NOC as a function of incidence angle), ascending orbits, data from 10 Dec - 19 Dec 2022: (a) NOCinc without HOC, (b) NOCinc after HOC.





**Figure 13.** C-band NOCant, ascending orbits, data from 10 Dec - 19 Dec 2022: (a) NOCant HH without HOC, (b) NOCant VV without HOC, (c) NOCant HH after HOC, (d) NOCant VV after HOC.



**Figure 14.** C-band wind speed/direction bias and their corresponding SDD as a function of wind speed/direction, data from 1 Dec to 10 Dec 2022, ascending orbits (a) NOCant; (b) HOC; (c) HOC&NOCant.

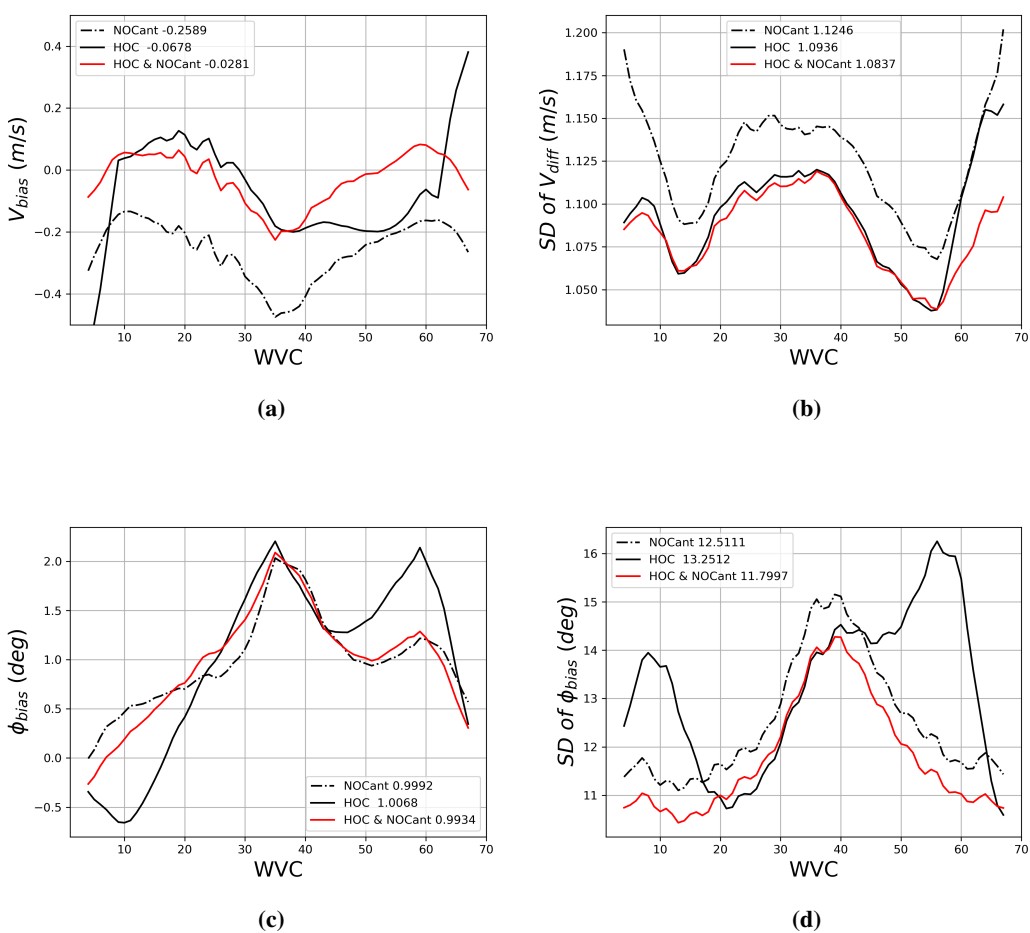

**Figure 15.** C-band wind retrieval statistics as a function of WVC with NOCant, HOC, and HOC&NOCant. The mean numbers of each condition are stated beside each legend, data from 1 Dec to 31 Dec 2022: (a) wind speed bias (b) wind speed SDD (c) wind direction bias (d) wind direction SDD.





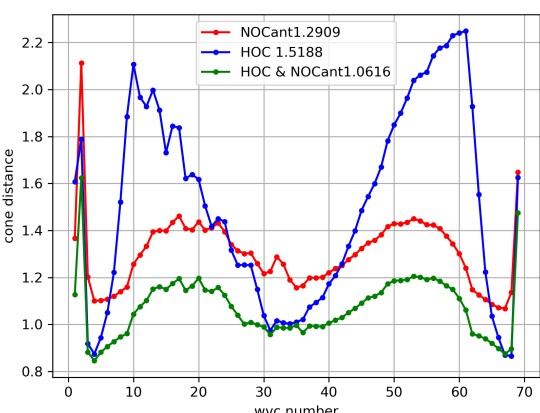

**Figure 16.** Average MLE (cone distance) as a function of WVC for C-band ascending orbits, NOCant is red, HOC is blue, HOC&NOCant is green.

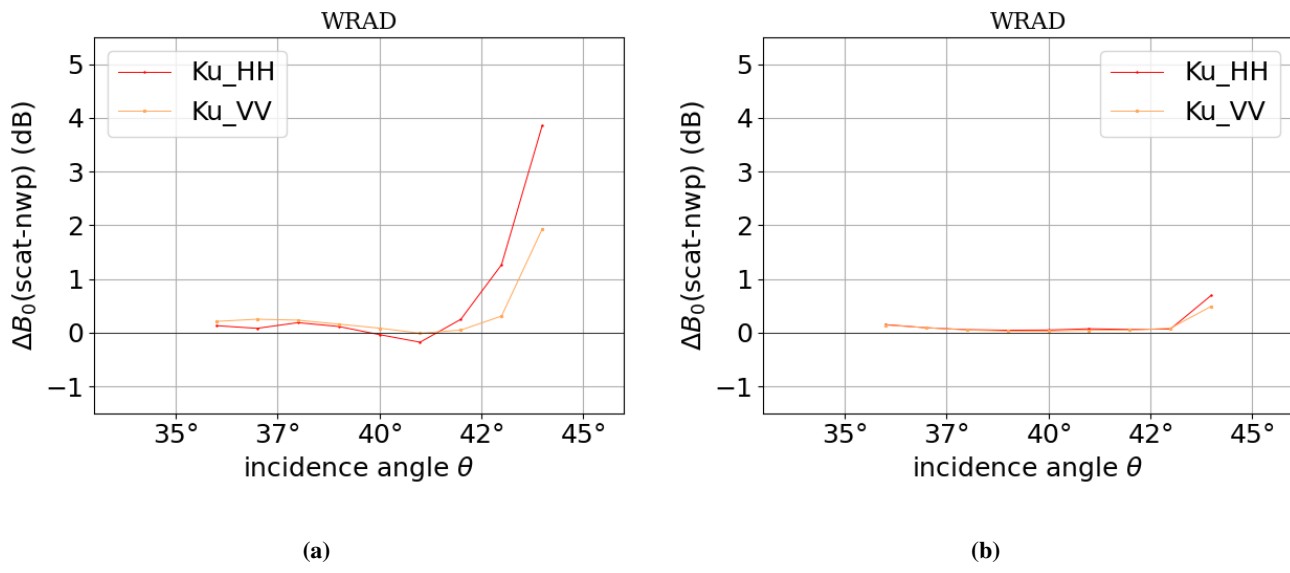

**Figure 17.** Ku-band NOCinc (NOC as a function of incidence angle), ascending orbits, data from 10 Dec - 19 Dec 2022: (a) NOCinc without HOC, (b) NOCinc after HOC.





**Figure 18.** Ku-band NOCant, ascending orbits, data from 10 Dec - 19 Dec 2022: (a) NOCant HH without HOC, (b) NOCant VV without HOC, (c) NOCant HH after HOC, (d) NOCant VV after HOC.



**Figure 19.** Ku-band wind speed/direction bias and their corresponding SDD as a function of wind speed/direction, data from 1 Dec to 10 Dec 2022, ascending orbits (a) NOCant; (b) HOC; (c) HOC&NOCant.



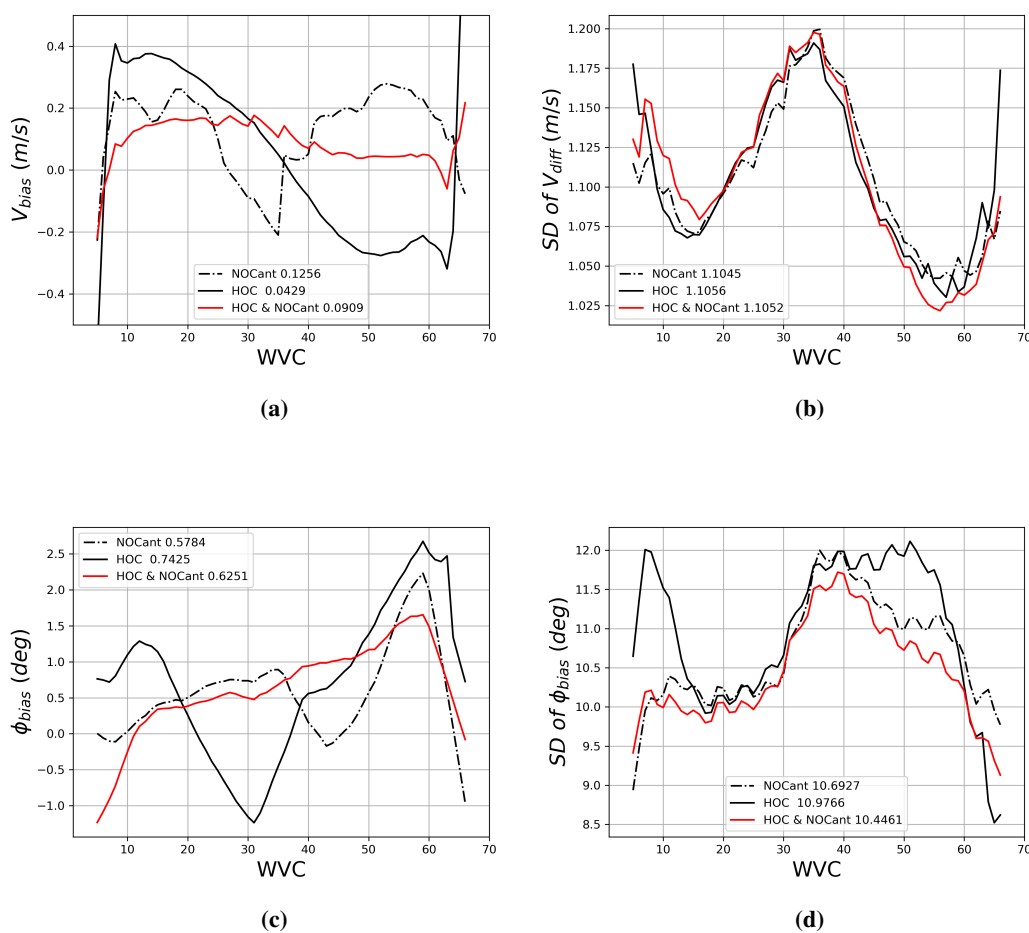

**Figure 20.** Ku-band wind retrieval statistics as a function of WVC with NOCant, HOC, and HOC&NOCant. The mean numbers of each condition are stated beside each legend, data from 1 Dec to 31 Dec 2022: (a) wind speed bias (b) wind speed SDD (c) wind direction bias (d) wind direction SDD.





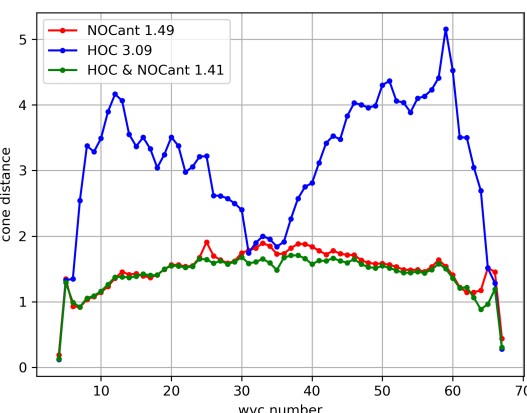

**Figure 21.** Average MLE (cone distance) as a function of WVC for Ku-band ascending orbits, NOCant is red, HOC is blue, HOC&NOCant is green.

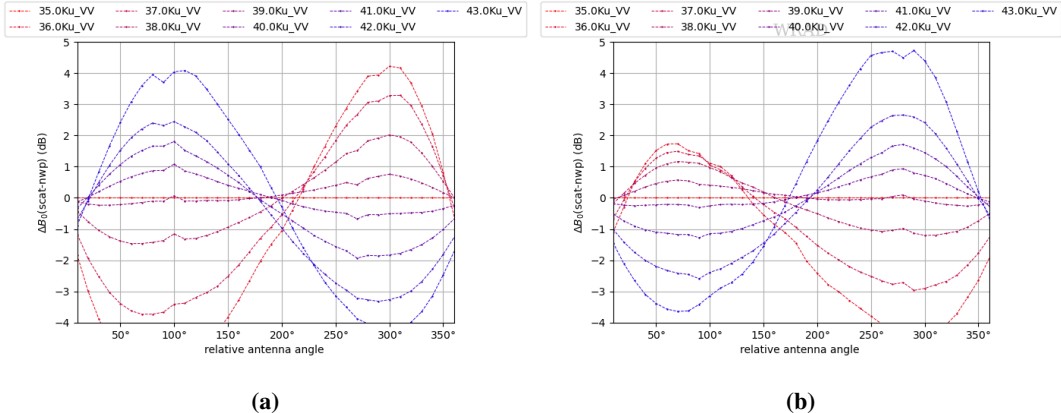

(a)                                                                    (b)

**Figure 22.** Ku-band NOCant (a) ascending orbits, (b) descending orbits.



*Code and data availability.* The wind processor WWDP can be aquired through https://scatterometer.knmi.nl/home/. Data can be acquired through the data distribution center of CMA: http://data.cma.cn/en, accessed on ** 2023.

*Author contributions.* Conceptualization, Z.L., A.S.; methodology, Z.L.; software, Z.L. and A.V.; validation, Z.L.; formal analysis, Z.L.; investigation, Z.L.; resources, Z.L.; data curation, Z.L. and A.V.; writing—original draft preparation, Z.L.; writing—review and editing, A.V., A.S.; visualization, Z.L.; supervision, A.S. and A.V.; project administration, Z.L., A.S. and A.V.; funding acquisition, A.S.. All authors have read and agreed to the published version of the manuscript.

*Competing interests.* At least one of the (co-)authors is a member of the editorial board of Atmospheric Measurement Techniques. The authors have also no other competing interests to declare.

*Acknowledgements.* We acknowledge the support of CMA in providing the data, status, and mission information. This research was funded by EUMETSAT OSI SAF CDOP4 (Darmstadt, Germany), the support from the EUMETSAT Ocean and Sea Ice Satellite Application Facility is also much appreciated. We thank Zhixiong Wang from Nanjing University of Information Science & Technology for his contribution to C-HH GMF.



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
