# Peer review of "Higher-Order Calibration on WindRAD scatterometer winds"

_Atmospheric Measurement Techniques, 2023_

## Author Comment (AC1)

**RC1**: 'Comment on amt-2023-112', Anonymous Referee #1

This manuscript proposes a higher-order calibration method for the WindRAD, with the objective of mitigating the non-linear characteristics of the radar measured sigma0, and in turn deriving high-quality winds. The methodology is well described, and the results are quite promising. I think the manuscript may draw common interests from the ocean surface wind community. However, a few minor questions need to be addressed before publication.

We would like to thank the reviewer for his/her valuable remarks which are helpful to improve the manuscript. All reviewer points are provided below in the black text, with our response directly beneath in blue.

1.      Do you use all of the sea surface data in the analysis of Figures 4 and 8? I would like to know how you deal with the quality control and the negative sigma0s.

Only the data with latitude between -55∘ to 65∘ are used, to exclude the sea ice. KNMI quality control has been applied, which excludes rain contamination and failed inversion, since the simulated NRCS in the calibration procedure assume a pure wind GMF. Negative sigma0s are not shown in the figures. Because of the noise, especially for Ku band (noisier than C band), there are negative sigma0s in the data. However, we cannot just throw out the negative sigma0s because they represent the low winds, and the climatological distribution of the wind might be distorted by throwing them out.  Negative sigma0s are used in the inversion by adding a penalizing logarithmic term in the cost function, derived from a Bayesian theorem assigning zero probability to negative winds. The resulting logarithmic term contributes strongly to the MLE, when the wind speed approaches zero and thus penalizes the cost function, in line with a low probability for low winds. It has been implemented in PenWP (Pencil-beam Wind Processor, Verhoef, 2018) and CWDP (CFOSAT Wind Data Processor, Li et al., 2021).

The text below is added in the manuscript at line 108:
'The σ∘ distributions shown in section 3.1 Fig. 4 and section 3.2 Fig. 8 are with a restriction on the latitude to be between 55∘S to 65∘N to exclude sea ice, and KNMI quality control has been applied, which excludes rain contamination and failed inversions. Negative σ∘s are not shown in the figures, while these are used.'

Verhoef, A. (2018). PenWP SoftwareEUMETSAT NWP SAF,Version 2.2. Retrieved from: https://nwp-saf.eumetsat.int/site/software/scatterometer/penwp/
Li, Z., Stoffelen, A., Verhoef, A., & Verspeek, J. (2021). Numerical weather prediction Ocean Calibration for the Chinese-French Oceanography Satellite wind scatterometer and wind retrieval evaluation. Earth and Space Science, 8, e2020EA001606. https://doi.org/10.1029/2020EA001606.

2.      The schematic illustration of HOC in Fig. 3 is fine. However, how do you calibrate each particular sigma0 value according the bias derived from Fig. 3? Is it done in linear space or in dB? Again, how do you deal with the negative sigma0s?

At the beginning of section 3 "HOC calculates σ∘ dependent calibration in intervals of 0.1 dB", which means for a particular sigma0 value, the bin for this sigma0 is determined first, then the HOC

calibration for this sigma0 bin is applied to the particular sigma0. The text is edited and moved to line 86 to make the explanation clearer:

'In practice, HOC calculates the σ∘ dependent calibration in intervals of 0.1 dB, using a lookup-table. First, the corresponding bin for a measured σ∘ is determined, and then the HOC calibration for this bin is applied to the measured σ∘.'

As explained in section 2.2, paragraph 2: "We take the CDF of the simulated σ∘ data as a reference and the CDF of the measured σ∘ data is calibrated in dB unit space with respect to the reference." The calibration is done in dB, and negative sigma0s are not included in HOC because the percentage of negative sigma0s is very low at about 0.4%, which implies that the CDF matching has very little shifting at the low sigma0s, and they are corresponding generally to the wind speed lower than 1m/s, therefore the impact would be minor. The way to utilize negative sigma0s is explained in the answer to the first comment.

Texts are added in the manuscript at line 88:
'Negative σ∘s are not included in HOC calculation because the percentage of negative σ∘s is very low at about 0.4%, which implies that the CDF matching has negligible shift at the low σ∘s, and they are generally corresponding to wind speeds lower than 1 m/s. Therefore the impact on the CDF matching is minor.'

3.      The azimuth-dependent bias in Figures 13 and 18 is quite suspicious. Does it exist in both versions of data? What could be the reason for the azimuth-dependent bias? Moreover, are the HOC and NOCant results the same for both versions of data?
Yes, it exists in both versions of data, the difference in the amplitude of the azimuth-dependent (wave pattern) is less for v2oper compared to v1oper.
There are two possible reasons: 1. The rotation might cause the antenna gain to decay and recover, thus leading to the azimuth-dependent variation; 2. The way level-0 data is derived. We do not know the exact method of how the level-0 data is derived by CMA (data provider); however, it relates to this wave pattern, because after the level-0 data update, the v2oper version has a reduced wave pattern. An azimuth-dependent calibration was analyzed for other rotating fan-beam scatterometers as well, such as CFOSAT (Li et al., 2021) and ISRO's ScatSat. The difference of the dependence between WindRAD and CFOSAT is that CFOSAT NOCant does not have such strong wave pattern, while ScatSat needed also an azimuth-dependent calibration correction.
The HOC and NOCant work on both versions of data, and HOC&NOCant gives the optimal calibration for both versions as well, which is explained in section 4.3.

Li, Z., Stoffelen, A., Verhoef, A., & Verspeek, J. (2021). Numerical weather prediction Ocean Calibration for the Chinese-French Oceanography Satellite wind scatterometer and wind retrieval evaluation. Earth and Space Science, 8, e2020EA001606. https://doi.org/10.1029/2020EA001606.

4.     It would be nice to see an illustration or a table on the HOC and the NOCant results.

The bin size of the HOC is 0.1dB, which means that there are 600 bins in total (dB range [-50, 10]). It is not very practical to show the HOC in a table, however, we show the HOC result in Figures 5 – 7 and Figures 9 – 11 for illustration purposes. Same reason for not showing NOCant in a table, but it is shown in Figures 13 and 18.

The wind retrieval results with HOC and NOCant are compared and shown in Figures 14, 15 and Figures 19, 20.

5.     Page 3, line 68: "correcting for air mass density" should be "correcting for the effect of air mass density"

It has been corrected.

---

## Author Comment (AC2)

**RC2**: 'Comment on amt-2023-112', Raj Kumar

This work is mainly to calibrate Fan Beam Scatterometer (WindRAD) data using proposed nonlinear calibration technique HOC. Manuscript is well written, however in my view it can be conscised more.

I have a few comments as following, which may require the explaination.

We would like to thank the reviewer for his valuable remarks which are helpful to improve the manuscript. All reviewer points are provided below in black text, with our response directly beneath in blue.

At many places in the manuscript, it has been mentioned that issues have been corrected in v2opr, so It is not clear, why V2opr has not been used for whole analysis of sigma0 to apply HOC based calibration, instead of using v1opr which has few drawbacks.

CMA (data provider) does not have a plan to reprocess v1oper, which means it is important that the proposed calibration methods (HOC and NOCant) work on both v1oper and v2oper, and v1oper has been available over a longer period than v2oper by far Therefore, v1oper is selected for doing the analysis.

Texts are added in the manuscript at line 63:

'CMA (China Meteorological Administration, the data provider) currently does not have a plan to reprocess v1oper, which means that it is important that the proposed calibration methods (HOC and NOCant) work on both versions v1oper and v2oper, where v1oper is available over a longer period than v2oper by far. Therefore, v1oper is selected for doing the main analysis.'

Line 127: Authors mention that nonlinearity can't be corrected by NOCant, however plots in the figure suggest that NOCant also may correct the data. One need to check the difference in both corrections.

Do you mean Figure 4 and 8? These figures show the comparison between the original sigma0 distribution and the HOC calibrated sigma0 distribution, where the non-linearity especially at low sigma0 values (asymmetric distribution along the diagonal) is calibrated by the HOC. However, NOCinc or NOCant can calibrate the ridge of the distribution onto the diagonal, but the non-linearity still exists, e.g., the figure below is the NOCant calibrated sigma0 distribution for C-band HH at incidence 38 deg. The ridge is on the diagonal, but the asymmetric distribution at low sigma0 values still exists.

[Figure]

Line 163: It will be also good to know the differences in wind retrieval for NOCint and HOC

Figure 13 is added to show the difference of the wind retrieval performance for NOCinc and HOC with the metric of MLE distance.

The text is edited as following at line 172:

"One metric to measure the quality of the wind retrieval is the MLE (also called cone distance) from equation (1). The MLE reveals how well the measurements fit the GMF, hence the smaller the MLE, the better the measurements fit the GMF, and the better wind retrieval can be expected. It is normalized by a WVC-dependent factor to obtain an expectation value of 1. This makes monitoring and quality control easier. Since different MLE normalizations lead to different outcomes, the same normalization and threshold are applied for all the calibration methods, such that the results can be directly compared. Fig. 13 shows the MLE metric comparison for NOCinc, HOC, and HOC&NOCinc. The MLEs for HOC and HOC&NOCinc are very close to each other, this implies that HOC is able to correct the incidence angle dependencies and the non-linear gain, hence the combination of HOC&NOCinc has a similar effect as HOC-only, as expected."

Figure 13a&b: What can be the reasons for HH & VV showing opposite behavior for low incidence angle.

The lowest incidence angles are not shown in Fig. 13, by adding the other two lower incidence angles in the figures (the plots below), it can be seen that it does not have the opposite behavior for low incidence angles, but the correction lifted entirely for lower incidence angles and the amount of the lifting is different between HH and VV. There is a larger distortion of the sigma0 distribution at lower incidence angles, which implies higher uncertainty and larger correction is needed. We excluded the very low and high incidence angles in the wind retrieval to avoid noise-dominated distributions.

[Figure]

Figure 13 c&d: After applying HOC correction, the correction doesn't seem to be uniform for all azimuth angles. For both HH&VV, correction seems to improve for azimuth directions above 180 degrees.

While we concluded that the HOC can shift the azimuth-dependent curve towards zero mean, it cannot remove the azimuth-dependent modulation, therefore, after the HOC correction, there is still azimuth-dependence in the data.

Like in the answer to the last comment, the two lower incidence angles are added in the plots (see the plots below). Lower incidence angles correspond to larger non-linearity in the sigma0 distributions, hence a larger HOC correction is applied, thus the NOCant for lower incidence angles are dragged down compared to no HOC applied for all the azimuth directions. The number of winds at the different azimuth angles is different, thus the weight for calculating NOCant values varies as a function of azimuth angles. This might lead to an non-uniform correlation for all azimuth angles.

[Figure]

Figure 14 suggest that NOCant doesn't improve the wind retrieval quality as directions retrieval accuracies are almost same. For wind speed also, there is only marginal improvement. Does it mean that even NOCint also would have performed in similar way.

The averaged wind speed bias value is indeed marginally improved for HOC&NOCant, but the figure shows wind speed bias as a function of wind speed: NOCant wind speed bias is from negative to positive for the wind speed from low to high. HOC improves the wind speed bias at lower wind speed, the wind speed bias as a function of wind speed is much flatter than for NOCant. HOC&NOCant further improve the wind direction statistics, therefore we conclude that a combined HOC&NOCant has the best performance.

The text is edited as follows at line 192:

'The wind speed biases are reduced when using either HOC or HOC&NOCant, which show a quite flat line, close to zero, as a function of wind speed, with only a small positive bias at high wind speeds remaining, probably due to poor sampling.'

NOCinc's performance is worse than NOCant, which is proven in Li et al., 2023, hence we did not include NOCinc in here.

Li, Z.; Verhoef, A.; Stoffelen, A.; Shang, J.; Dou, F. First Results from the WindRAD Scatterometer on Board FY-3E: Data Analysis, Calibration and Wind Retrieval Evaluation. Remote Sens. 2023, 15, 2087. https://doi.org/10.3390/15082087

Figure 15: It is not clear, why HOC should be higher for sweet spot (centre WVCs). Normally at these WVCs, SD should be low.

The center WVCs are located at the nadir swath, which means the geometry (azimuth angle) is either looking forward or looking backward, so they are not sweet spots. The WVCs located between the outer and nadir swath are the sweet WVCs. The diversity of the geometry for center WVCs is limited and the sensitivity to the wind vector rather variable, therefore, the SD is higher at the center WVCs.

Line 214: Authors have made a good point that MLE is lowest for HOC with NOCant, whereas the difference between NOCant alone and HOC combined with NOCant is almost same. What can be the reason that it is unlike C band. The nonlinearlty in the sigma0 should be frequency dependent for a particular angle (or range of angles) only.

For Ku-band, the incidence angles between 38° and 41° (medium angles) are selected for the wind retrieval. For C-band, the incidence angles between 36° and 43° (medium angles) are selected. The low and high incidence angles are excluded for both C- and Ku-band. As shown in Figure 4 (b, e) and Figure 8 (b, e), which represent the sigma0 distribution of medium incidence angles for C-band and Ku-band respectively, the non-linearity is much stronger for C-band than for Ku-band. Therefore, HOC calibration has much higher positive influence on C-band than on Ku-band, which leads to much less differences between NOCant alone and HOC&NOCant for Ku-band.

The text in the manuscript have been adjusted as follows at line 224:

"but the difference between NOCant and HOC&NOCant is smaller than for C-band, probably because the non-linearity in the σ° distribution for the used incidence angles (38° to 41°, which are in the medium range, represented in Fig. 8(b) (e)) is much smaller for Ku-band than for C-band (Fig. 4(b) (e))."

Line 220: Couldn't understand the mirroring effect between ascending and ascending. It will be advisable to explain it.

In the process of checking the operational routines for WindRAD, a bug in unit conversion was found in data preprocessing (level 0 to level 1), which causes the phenomena described in the text. After fixing the bug, new calibration factors were calculated for C and Ku (HH and VV polarization) and have been implemented in the ground system. The data from 22 April 2023 is the new version. Changes and analysis are reported in an NSMC internal report: "Technical report on operational software change application of FY-3E WindRAD". The announcement of the version update was published on the NSMC internal website.

The following text has been added in the manuscript at line 232:

"The red dashed lines of the relative antenna azimuth in ascending orbits are negative between 0° and 200°, whereas they are positive between 200° and 360° (Fig. 22 a). For descending orbits, the signs are opposite (Fig. 22 b)."

"These puzzling phenomena were caused by a unit conversion bug in the data preprocessing from level-0 to level-1 and they are corrected in the v2oper data. New calibration factors for level-0 and level-1 data were also calculated for both C-band and Ku-band (HH and VV polarization), which have been implemented in the ground system. The analysis is reported in NSMC (National Satellite Meteorological Center) internal report (Shang et al., (2023))."